# Pituitary Tumor-Transforming Gene 1/Delta like Non-Canonical Notch Ligand 1 Signaling in Chronic Liver Diseases

**DOI:** 10.3390/ijms23136897

**Published:** 2022-06-21

**Authors:** Meritxell Perramón, Wladimiro Jiménez

**Affiliations:** 1Biochemistry and Molecular Genetics Service, Hospital Clínic Universitari, 08036 Barcelona, Spain; mperramon@clinic.cat; 2Institut d’Investigacions Biomèdiques August Pi i Sunyer (IDIBAPS), Centro de Investigación Biomédica en Red de Enfermedades Hepáticas y Digestivas (CIBEReHD), 08036 Barcelona, Spain; 3Department of Biomedicine, University of Barcelona, 08036 Barcelona, Spain

**Keywords:** chronic liver diseases, pituitary transforming gene 1, delta like non-canonical notch ligand 1, hepatic steatosis, hepatic fibrosis, hepatocarcinogenesis

## Abstract

The management of chronic liver diseases (CLDs) remains a challenge, and identifying effective treatments is a major unmet medical need. In the current review we focus on the pituitary tumor transforming gene (PTTG1)/delta like non-canonical notch ligand 1 (DLK1) axis as a potential therapeutic target to attenuate the progression of these pathological conditions. PTTG1 is a proto-oncogene involved in proliferation and metabolism. PTTG1 expression has been related to inflammation, angiogenesis, and fibrogenesis in cancer and experimental fibrosis. On the other hand, DLK1 has been identified as one of the most abundantly expressed PTTG1 targets in adipose tissue and has shown to contribute to hepatic fibrosis by promoting the activation of hepatic stellate cells. Here, we extensively analyze the increasing amount of information pointing to the PTTG1/DLK1 signaling pathway as an important player in the regulation of these disturbances. These data prompted us to hypothesize that activation of the PTTG1/DLK1 axis is a key factor upregulating the tissue remodeling mechanisms characteristic of CLDs. Therefore, disruption of this signaling pathway could be useful in the therapeutic management of CLDs.

## 1. Introduction

Liver diseases have become a first-order health problem in Western countries. The burden of these diseases in Europe is the largest in the world and continues to grow [1]. Regardless of their etiopathogenic origin, advanced liver diseases display several common traits. Among these, perhaps the most characteristic is the development of liver fibrosis, leading to severe consequences including cirrhosis and hepatocellular carcinoma (HCC).

Fibrosis results from chronic injury of the hepatic parenchyma and consists of very active extracellular matrix remodeling with progressive and abundant deposition of collagen fibers. During the process of hepatic fibrogenesis there is a complex interrelation between the different cell types present in the liver. Following liver injury, damaged hepatocytes release reactive oxygen species (ROS) and other fibrogenic mediators that induce the recruitment of inflammatory cells. Activated hepatic stellate cells (HSC) play a major role in the pathogenesis of fibroproliferative processes. Among HSC activation promoters there are various growth factors such as transforming growth factor-β (TGFβ) and platelet derived growth factor (PDGF), as well as vasoactive substances (thrombin, angiotensin II, and endothelin-1), cytokines (monocyte chemotactic protein-1), and adipokines (leptin) [2]. In addition, the activation and proliferation of HSC is regulated through molecular mechanisms that include various intracellular signaling pathways (mitogen-activated protein kinase (MAPK), phosphatidylinositol 3-kinase (PI3K)/AKT, peroxisome proliferator-activated receptor gamma, nuclear factor kappa-light-chain-enhancer of activated B cells, and toll-like receptors). Recently, a new player has been identified in this context—the pituitary tumor transforming gene 1 (PTTG1)/delta like non-canonical notch ligand 1 (DLK1) signaling pathway. The present article intends to extensively review the major mechanisms regulating synthesis, secretion, and maturation of PTTG1 and DLK1, as well as the principal evidence pointing to these peptides as possible antifibrogenic targets for the treatment of advanced liver disease.

### 1.1. PTTG1: Gene Structure and Regulation, Protein Synthesis, and Maturation and Tissue Expression

PTTG1, also named tumor-transforming protein 1, was first isolated from rat pituitary tumor cells by Pei and Melmed in 1997 [3]. PTTG1 is the index mammalian securin, a key mediator involved in faithful sister chromatid separation during mitosis. Furthermore, PTTG1 overexpression is involved in cellular transformation aneuploidy and has been associated with highly proliferative tumors.

Human *PTTG1* (*hPTTG1*) is located on chromosome 5 (5q33.1) and consists of six exons and five introns [4]. This gene has homologs across species such as rats, mice, and chickens. In rats, it is located on chromosome 10 (10q21) and shares 85% homology with the *hPTTG1* DNA sequence, whereas in mice it is located on chromosome 11 and shares 78% homology [5]. Three transcript variants encode the same protein.

Two homologous genes, *PTTG2* and *PTTG3*, with unknown physiological function, have been identified. They are intronless and in humans they are localized on chromosome 8q13.1 and 4p15.1, respectively [6].

*hPTTG1* has two transcription initiation sites, one at −37 and the other at −317 bp upstream of the ATG translation start site [4,7]. In addition, this gene has an enhancer element between −706 and −407 bp, three SP1/GC boxes, three AP1 and one AP2 binding site, and a cyclic AMP and an insulin response element [7]. SP1 and NF-Y transcription factors are needed for *PTTG1* promoter activity and their directed mutagenesis results in reducting in promoter activity of 70 and 25%, respectively [4]. In contrast, the transcription factor and tumor suppressor Krüppel-like factor 6 (KLF6) has been found to act as a repressor by binding between −406 and −246 bp of the *PTTG1* promoter [8,9].

PTTG1 is a highly hydrophilic multidomain protein located mainly in the cytoplasm as well as in the nucleus [10]. PTTG1 has a functional C-terminal and a regulatory N-terminal domain. Near the C-terminal end, PTTG1 has a proline rich region proper for the SH3-binding motif (58–101 amino acids (aa)), a transactivating domain (119–164 aa) and a PTTG-binding factor (PBF) binding region [5,11]. Close to the N-terminal region, PTTG1 has a DNA binding region (61–118 aa), a destructive box (D-box, 61–68 aa), a SH3- binding motif (51 and 54 aa), and a KEN Box (9–11 aa) (Figure 1). D- and KEN boxes regulate PTTG1 degradation [5,10].

PTTG1 activity is mostly regulated at the posttranscriptional level by phosphorylation. During mitosis, cyclin-dependent kinase (Cdk-2) phosphorylates PTTG1 at Ser^165^ [12]. PTTG1 can also be phosphorylated by MAPK at Ser^162^ in the consensus motif Pro-X-(Ser/Thr)-Pro to induce nuclear translocation and transactivation activity [11]. Additionally, DNA-dependent protein kinase (DNA-PK) also phosphorylates PTTG1, although the exact location is unknown [13].

PTTG1 is highly expressed in fetal liver and the placenta [14]. In contrast, PTTG1 expression in adult healthy tissue is restricted to the testis and thymus [12]. Weak expression has been detected in the colon, small intestine, spleen, brain, and pancreas [5]. PTTG1 levels are elevated in highly proliferative cells during the cell cycle, especially in the G_2_/M phase of mitosis [15].

PTTG1 is an oncogene involved in cellular transformation and aneuploidy [10]. Accordingly, PTTG1 is highly expressed in endocrine and non-endocrine malignancies such as pituitary adenoma, astrocytoma, thyroid cancer, breast and ovarian cancer, renal cell carcinoma, lung cancer, colorectal carcinoma, HCC, and leukemia [10,16,17,18,19,20].

### 1.2. Biological Functions of PTTG1

PTTG1 is involved in a myriad of functions, including cell cycle and gene regulation, DNA repair, metabolism, and organ development. Mice knockout for the *Pttg1* gene are viable and fertile but display thymic hyperplasia, thrombocytopenia, testicular and splenic hypoplasia, aberrant fibroblast cell cycle progression, and premature centromere division [21]. Additionally, these mice present male-associated diabetes in late adulthood. Their beta cells show abnormal cycle progression and as a result these mice exhibit hyperglycemia and reduced insulin levels [22,23]. Spermatocytes and spermatids forming the testis of healthy adults express PTTG1, suggesting a role in male germ cell differentiation [12].

PTTG1 is a securin that regulates chromatid separation in the course of mitosis by inhibiting the activity of separase and maintaining sister chromatids together until the end of the metaphase [24]. During cell proliferation, Cdk-2 phosphorylates PTTG1 at Ser^165^. Interestingly, this phosphorylation site is located in the proline-rich region, which is important for binding with proteins that contain SH3- motif [12]. At the onset of the anaphase, the anaphase-promoting complex ubiquitinates PTTG1 to mediate its degradation, which is D box and KEN box-dependent [10,15].

Epidermal growth factor (EGF), a well-known mediator involved in proliferation, has been described as an inducer of PTTG1 expression via PI3K, protein kinase C, and MAPK pathways [25]. PTTG1 interacts with MEK1 through the SH3- binding motif, which is required to mediate MAPK phosphorylation of PTTG1 at Ser^162^ and the activation of its transactivation function. The consensus phosphorylation site for MAPK overlaps to that of Cdk-2 [11]. Once phosphorylated and activated, PBF enhances PTTG1 translocation into the nucleus, where PTTG1 can regulate gene expression binding to a myriad of transcriptional factors and up to 700 promoters [26]. Interestingly, insulin-like growth factor 1 (IGF-1) and insulin also activate PI3K/AKT signaling to induce PTTG1 expression [27]. Once in the nucleus, PTTG1 promotes cyclin D3 expression together with the transcription factor Sp1 to induce G_1_/S phase transition and represses the p21 promoter mediating cell senescence [28]. Other well-established targets include fibroblast growth factor 2 (FGF2) and c-Myc. FGF2 is a mediator involved in endothelial cell proliferation and survival, as well as angiogenesis. In the presence of PBF, PTTG1 binds to the FGF2 promoter to induce its transcription [26]. Similar effects result from the PTTG1 complex with upstream stimulatory factor binding in the c-Myc promoter (Figure 2) [29]. c-Myc is a protooncogene participating in the regulation of genes related to important cellular processes such as differentiation, proliferation, apoptosis, angiogenesis, metabolism, immune response, and protein translation [30]. PTTG1 is also involved in angiogenesis by regulating several mediators such as vascular endothelial growth factor (VEGF) and metalloproteinase 2 (MMP2) [31].

PTTG1 also interacts with the protein Ku-70, the regulatory subunit of DNA-PK. Upon DNA damage such as double-strand breakage, Ku-70 releases PTTG1 which is then phosphorylated by DNA-PK to block cell cycle progression thus maintaining chromosomal stability [13]. PTTG1 has also been found to interact with thyroid hormone β receptor, which mediates PTTG1 proteosomal degradation when attached to thyroid hormone 3 (T3) [32]. Moreover, Chen et al. suggested that T3 indirectly inhibits PTTG1 transcriptional activity via Sp1 [33]. PTTG1 is also able to prevent apoptosis by direct interaction with p53 [34] and suppress prolactin synthesis [35].

Finally, PTTG1 is considered an oncogene and its overexpression has been related to cell growth, epithelial mesenchymal transition, angiogenesis, invasion, and metastasis in cancer [36].

### 1.3. DLK1: Gene Organization and Regulation, Protein Structure, and Expression in Different Tissues

*DLK1*, also known as preadipocyte factor-1 or fetal antigen-1, is a member of the Notch/Delta/Serrate family [37]. It was first cloned from mice in 1993 by Smas and Sul [38]. Almost simultaneously, Laborda et al. isolated and characterized the human and mouse cDNA [39]. The most well-known function of DLK1 is as an adipogenesis inhibitor. During development, *DLK1* is broadly expressed, but in adult tissues it is restricted to several organs and cellular subsets. *DLK1* is overexpressed in several cancers and is related to bad prognoses. The human *DLK1* gene consists of five exons and four introns, and two transcript variants have been described [40]. It is encoded in the *DLK1- type III iodothyronine deiodinase (DIO3)* gene cluster located in chromosome 14 (14q32.2). *DLK1*-*DIO3* homologs have been found in all vertebrates and their organization and imprinting is highly conserved among humans, mice, and sheep [41,42]. *Dlk1* is located in chromosomes 6 (6q32) and 12 (12qF1) of rats and mice, respectively.

The *DLK1*-*DIO3* cluster contains three protein coding genes (*DLK1*, *DIO3*, and retrotransposon-like 1 (*RTL1*) expressed paternally, in addition to long non-coding RNAs *(MEG3*, *MEG8*, *MEG9*, and *LINC00524*), two families of small nucleolar RNAs (*SNORD113* and *SNORD114*), two large clusters of micro RNAs (miRNAs), and several pseudogenes are expressed maternally [43]. Although the exact mechanisms involved in *DLK1**-DIO3* imprinting regulation are not fully known, three major imprinting control regions with differential methylation of maternal and paternal alleles have been described. The first imprinting control region is an intragenic differentially methylated region (DMR) situated 13 bp upstream to the MEG3 transcription start site, while the second is a DMR located in the promoter of MEG3 and the latter in the second intron of MEG8 (Figure 3) [44,45]. Additionally, reciprocally imprinted domains have also been shown to interfere with the expression of products of both parental chromosomes [46].

Furthermore, the DLK1 promoter has several CpG dyads, and their methylation may also contribute to regulate *DLK1* gene expression. It has been reported that KLF6 binding with histone deacetylase 3 to the *DLK1* promoter (between −78 and −53 bp upstream of the transcriptional start site) results in *DLK1* repression [47]. Contrarily, the E2F Transcription Factor 1 has been shown to activate the DLK1 promoter, and its activity is inhibited by a tripartite motif-containing protein 28 [48]. Finally, at the proximal 5′ enhancer region, the *DLK1* gene includes three putative hypoxia responsive elements at −758, −402, and −248 bp, and consequently its expression is induced by hypoxia inducible factor 1α (HIF-1α) and HIF-2α [49].

The *DLK1* gene encodes for a single pass transmembrane protein of 383 aa. The DLK1 protein consists of a signal peptide (1–23 aa), an extracellular domain of six tandem EGF repeats (24–55, 53–86, 88–125, 127–168, 170–206, 208–245 aa), a juxtamembrane region, a transmembrane domain (304–327 aa), and a short cytoplasmic tail (328–383 aa) (Figure 4) [50]. These EGF repeats are characteristic of proteins related to cell growth and differentiation such as tumor growth factor alpha. However, due to the lack of certain conserved residues, DLK1 cannot bind to the EGF receptor [51]. DLK1 is a member of the Notch/Delta/Serrate family and shares high structural homology with this family. Nonetheless, DLK1 lacks the delta/serrate/LAG-2 (DSL) domain that mediates receptor–ligand interaction that is characteristic of all Notch ligands, and thus it acts as an antagonist of Notch signaling [51,52].

The tumor necrosis factor alpha converting enzyme (TACE), also named A disintegrin and metalloproteinase 17, mediates the proteolytic cleavage of DLK1 into two soluble forms [53,54]. Shedding near to the fourth EGF repeat results in a large 50 kDa product, whereas the cleavage in the region proximal to the transmembrane domain produces a small 25 kDa form. The activity of this small form remains controversial, but some evidence indicates that only the larger form is biologically active [55].

The human DLK1 has two alternate splicing isoforms: one tethered to the membrane with autocrine effects, and the other cleavable with juxtacrine actions. In contrast, mice DLK1 has six isoforms, with A and B able to produce the larger soluble form and C, C2, D, and D2 being membrane-bound isoforms [50]. All isoforms can yield the smaller soluble product.

During embryonic development, DLK1 is detected in all three embryonic layers and its expression decreases as differentiation increases [52]. DLK1 has been identified in cells from endoderm-derived tissues, such as prostate, pancreas, liver, and lung; in mesoderm-derived tissues including the heart, cartilage, bone, skeletal muscle, brown adipose tissue, kidney, testis, and ovary, as well as ectoderm-derived structures including the pituitary gland, pons, and brain ventricle and mixed origin tissues such as the adrenal gland [34,38,39]. DLK1 is overexpressed in the yolk sack and placenta and elevated circulating levels are found in maternal blood during late pregnancy, the fetus being the primary source [56,57].

In contrast, in adults DLK1 expression is generally restricted to neuroendocrine tissues and stem progenitor cells, including the zona glomerulosa of the adrenal gland, pituitary gland, insulin-producing β cells in the pancreas, basal layer of epithelial cells in the prostate, theca interna cells of the ovary, precursor Leydig cells of the testes, epithelial cells of the proximal tubules in the kidney, preadipocytes, and some neurons in the mesencephalon and pons [50].

DLK1 dysregulation has been documented in several diseases such as metabolic disorders and muscle, lung, and liver regeneration [58,59]. In cancer, DLK1 acts as an oncogene, and elevated expression has been reported in endocrine and non-endocrine malignancies including prostate and ovarian cancers, neuroblastoma, pancreas and colon adenocarcinomas, non-small and small cell lung cancer, HCC, and Wilm’s tumor [50].

### 1.4. Biological Functions of DLK1

DLK1 is implicated in various differentiation processes such as osteogenesis, neurogenesis, hematopoiesis, and differentiation of hepatocytes. In fact, mice lacking *Dlk1* show growth retardation, elevated adipose mass and serum lipids, skeletal and eyelid deformations [60], decreased bone mass [61], alterations in B cell development [62], and elevated serum leptin and decreased follicle-stimulating hormone [63]. Despite the lack of the DLS motif needed for canonical ligand interactions with Notch, soluble DLK1 has been shown to induce both Notch-dependent and independent signaling pathways.

DLK1 stimulates oscteoclastogenesis inducing bone resorption and inhibiting bone formation by progenitor cell inflammatory cytokine production [64]. Moreover, DLK1 interacts with the cysteine rich FGF receptor and impairs FGF18 binding and the consequent intracellular signaling pathway, which is important for chondrocyte and osteoblast development [65]. DLK1 also promotes cycle exit of neural progenitors to form neurons via suppression of SMAD activation when challenged with bone morphogenetic proteins [66]. During hematopoiesis, DLK1 inhibits stem cell factor-induced colony formation of hematopoietic progenitors [67]. In addition, DLK1 is highly upregulated in the hepatoblasts of the embryonic liver, conferring a proliferative phenotype and the ability to differentiate in hepatocytes or cholangiocytes [68].

However, the best-known function of DLK1 is inhibition of adipogenic differentiation. The preadipocyte cell line 3T3-L1 constitutively expresses DLK1, which decreases after differentiation to mature adipocytes. This process is prevented by adding the 50 kDa soluble form [54]. Transgenic mice overexpressing this large form show reduced body weight and an impairment in adipocyte differentiation with smaller fat pads, including brown fat, and reduction in the number of adipocytes and adipogenic factors released [69]. DLK1 interacts with fibronectin and together they activate the extracellular signal-regulated kinase (ERK)/MAPK pathway, upregulating Sox9 which binds to C/EBPβ and C/EBPδ promoters to suppress their activity and inhibit preadipocyte differentiation [70,71]. Besides, DLK1 can also modulate adipocytes by inhibiting the activity of each one of the four mammalian Notch with a feed-back mechanism [72]. Despite most findings suggesting that only the larger soluble form has an anti-adipogenic activity [53,73], some authors have suggested that membrane-tethered DLK1 could reduce preadipocyte proliferation by inhibiting G_1_ to S phase transition [74]. A membrane-bound non-EGF-like DLK1 region has also been found to interact with extracellular IGF binding protein 1 and trigger the adipogenesis of 3T3-L1 cells by releasing IGF-1 and enhancing IGF receptor signaling [75]. Some evidence also suggests that growth hormone (GH) blocks predipocyte differentiation by increasing forkhead box protein A2 (Foxa-2) expression. Foxa-2 acts as a direct transcriptional activator of DLK1 by binding to its promoter [76]. Additionally, glucocorticoids can downregulate DLK1 [77] and DLK1 can bind to the GH receptor to repress its expression [61]. Finally, DLK1 expression is enhanced under hypoxia by HIF-1α and/or HIF-2α binding in the proximal 5′ promoter region of the *DLK1* gene [49]. Recent findings also suggest that HIF-1α and or HIF-2α also activate TACE proteolytic cleavage resulting in the release of the intracellular DLK1 fragment (Figure 5) [78]. DLK1 can therefore act as an autocrine, a paracrine, or an endocrine mediator.

Finally, since DLK1 is able to maintain cells in an undifferentiated phenotype and its overexpression is related to augmented proliferation, angiogenesis, and epithelial to mesenchymal transition (EMT), all of which are hallmarks of cancer, and its dysregulation is implicated in the development of several diseases and malignancies. The expression of DLK1 and PTTG1 transcripts correlates with several normal and tumor tissues, including normal pituitary tissue and samples of pituitary adenoma or fetal liver and HCC. Interestingly, Espina et al. also reported that PTTG1 dramatically induces DLK1 expression independently of Foxa-2 in preadipocytes inhibiting adipocyte differentiation [14]. Adipocyte maturation and liver HSC transdifferentiation share common regulatory mechanisms related to adipogenic transcription regulation. First, quiescent HSC are lipid-storing pericytes that abundantly express genes related to adipogenic transcription [79]. Secondly, both preadipocytes and quiescent HSC are large lipid stores, which are lost following transdifferentiation into mature adipocytes or activated HSC. Thirdly, profibrogenic mediators such as PDGF are known to suppress adipocyte differentiation in both types of cells. Finally, immature adipocytes and quiescent HSC express extracellular basement membrane matrix proteins, which are replaced by interstitial collagens in both activated cells [80]. For all these reasons, PTTG1/DLK1 axis could be of major importance in the pathogenesis of liver diseases.

## 2. Involvement of PTTG1 and DLK1 in the Pathogenesis of Chronic Liver Diseases (CLDs)

The global burden of CDLs is rising every year. However, its prevalence varies among regions largely due to differences in the incidence of risk factors including obesity, viral hepatitis, and alcohol consumption. Therefore, demographic and historical factors need to be taken into account when trying to understand the epidemiology of CDLs [1].

### 2.1. Non-Alcoholic Fatty Liver Disease (NAFLD)

NAFLD is currently the most common cause of chronic liver disease in the world, with a prevalence of 20–40% in the general population and up to 95% in subjects with obesity and diabetes [81,82]. NAFLD is characterized by abnormal accumulation of fatty acids (FAs) and the presence of steatosis in more than 5% of hepatocytes [83]. This disease includes a wide spectrum of liver diseases, ranging from asymptomatic simple fatty liver to potentially irreversible non-alcoholic steatohepatitis (NASH). NASH is characterized by the accumulation of hepatocellular lipids and is distinguished from simple steatosis by the presence of hepatic inflammation, oxidative stress, apoptosis, and fiber. NASH frequently progresses to fibrosis, cirrhosis, HCC, and liver failure [84,85,86,87].

Fat accumulation in the liver occurs as a result of a disturbance in the balance between FA uptake, synthesis, export, and oxidation [88]. FA are stored in adipose tissue in the form of triglycerides (TG). However, in obese or insulin-resistant subjects, FAs appear to be diverted from their primary storage site to ectopic sites such as skeletal and liver tissues, possibly through increased adipocyte lipolysis. The uptake of free FAs by these organs is facilitated by FA transport proteins and CD36 FA translocase, the expression of which is elevated in obese patients and in patients with NAFLD [89,90]. Once they reach the liver, these FAs can be oxidized by β-oxidation or they can be esterified in the form of TG, accumulating in liver tissue [91].

Among the multiple mechanisms involved in the development of NAFLD, insulin resistance seems to play an essential role [92,93]. Under conditions of insulin resistance, insulin does not adequately inhibit hormone-sensitive lipase and, thus, lipolysis in white adipose tissue is not suppressed. Therefore, peripheral fats stored in adipose tissue reach the liver through the circulation system as non-esterified fatty acids. Fat from the diet is also absorbed by the liver through the absorption of chylomicrons derived from the intestine. In addition, the combination of elevated plasma glucose levels (hyperglycemia) and high insulin concentrations (hyperinsulinemia) promote hepatic de novo FA lipogenesis and impair β-oxidation, thereby contributing to the development of hepatic steatosis [94]. After the esterification stage (conversion of FA to TG), TG can be stored within hepatocytes in the form of lipid droplets or secreted into the blood in the form of very low-density lipoproteins.

The increase in the hepatic content of lipids results in higher sensitivity to the different factors causing hepatocyte damage and, therefore, progression to more severe stages of the disease. Likewise, insulin resistance also promotes adipose tissue dysfunction with the consequent alteration in the production and secretion of adipokines and inflammatory cytokines [95]. The excess of fat accumulated in the liver in the form of TG, together with the lipotoxicity produced by the high levels of free FA, free cholesterol, and other lipid metabolites, causes mitochondrial dysfunction accompanied by oxidative stress, production of ROS, and activation of the mechanisms associated with the stress of the endoplasmic reticulum [96]. In addition, there are several indirect evidence suggesting that the PTTG1/DLK1 axis could also be involved in the pathogenesis of NAFLD.

Early investigations by Espina et al. [14] in 2009 already showed that DLK1 is one of the most abundantly expressed PTTG1 targets. DLK1 participates in several differentiation processes, including adipogenesis. By forcing PTTG1 gene expression in 3T3-L1 cell lines, these authors demonstrated that this gene inhibits adipogenesis, thus promoting DLK1 RNA stability and accumulation. These findings indicate that PTTG1 participates in adipocyte differentiation by regulating the DLK1 gene. These results recalled attention to previous studies demonstrating that the hepatic expression of a full extracellular preadipocyte domain of factor 1 (namely DLK1) gene decreased adipose tissue mass in mice [69]. Additionally, Villena et al. [97] also showed that transgenic mice overexpressing DLK1 were protected to diet-induced obesity but were insulin resistant and displayed increased levels of circulating lipids. Results from Kavalkova P et al. [98] supported the role of DLK1 in insulin resistance. They found that normal-weight or obese female human subjects did not show differences in serum DLK1 concentrations, whereas females with obesity and type 2 diabetes mellitus significantly augmented circulating DLK1 levels compared to normal-weight subjects. Furthermore, more recently Charambourus et al. [99] showed that the increase in DLK1 gene expression results in reduced fat stores, IGF-1, and a defect in feedback regulation of GH, which finally leads to a reduction of hepatic steatosis. However, the exact role of DLK1 in the progression of NAFLD remains elusive and controversial. Recently, Jensen CH et al. [100] reported that elevated DLK1 circulating levels were associated with increased body fat and the development of metabolic syndrome in humans. They also showed that DLK1 deficiency protected against obesity and insulin resistance by negatively regulating GLUT4-mediated skeletal muscle glucose uptake in a high-fat diet experimental mice model.

Accumulating evidence also suggests that DLK1 involvement in the pathogenesis of NASH could be partially explained by the antagonism of Notch signaling. Whereas Notch activity is absent in adult hepatocytes, it is augmented in the liver of patients with NASH and in experimental models of diet-induced NASH. In these conditions, Notch upregulation enhances hepatocyte gluconeogenesis and insulin resistance by inducing the transcription of the catalytic subunit of glucose-6-phosphatase and phosphoenolpyruvate carboxykinase in a FoxO1 dependent manner [101]. In line with these observations, Lee YH et al. [102] demonstrated that exogenous administration of DLK1 reduced steatosis and hyperglycemia via 5′ AMP-activated protein kinase activation in the liver of db/db mice. These authors also showed that DLK1 administration suppressed hepatic glucose production by enhancing Akt phosphorylation and subsequent nucleus translocation of Foxo1. Furthermore, gene structural analysis has revealed that, by directly targeting DLK1, different miRNA represses its expression, resulting in hepatic steatosis [103]. Moreover, it has been shown that MiR-124-3p expression in the liver is increased during high fat diet intake. MiR-124-3p controls DLK1 transcription unveiling MiR-124-3p as a potential target for clinical interventions of NAFLD. In summary, there are compelling data strongly suggesting that dysregulation of the PTTG1/DLK1 axis is involved in the pathogenesis of fat metabolism in liver diseases.

### 2.2. Fibrosis

Liver fibrosis is the wound-healing homeostatic response resulting from repeated hepatic injury [104]. After acute liver injury, parenchymal cells regenerate and replace necrotic or apoptotic cells. This process is associated with inflammatory response and limited extracellular matrix (ECM) deposition. If liver injury persists, liver regeneration eventually fails, and hepatocytes are replaced by an excess of ECM, mostly collagen fibers. As liver disease progresses, the fibrogenic process becomes chronic and leads to severe fibrosis and finally cirrhosis [105]. Liver fibrosis is associated with significant alterations in both the quantity and composition of the ECM. In advanced stages of fibrosis, the liver contains approximately 6 times more ECM than normal. The composition of the ECM mainly includes collagens I, III, and IV, fibronectin, undulin, tenascin, elastin, laminin, hyaluronic acid, and proteoglycans. The accumulation of ECM is the result of increased synthesis and impaired degradation [106]. The decrease in the activity of the MMPs that degrade the ECM is mainly due to an overexpression of their specific inhibitors, the tissue inhibitors of metalloproteinases (TIMPs) [105].

HSCs are the main ECM-producing cells in injured liver [107]. In the healthy liver, HSCs are located in the space of Disse, between the basolateral surface of the hepatocytes and the antiluminal face of sinusoidal endothelial cells. They account for 15% of the cell population in healthy livers and their main function is storage of vitamin A and other retinoids [108]. After chronic liver injury, HSCs are activated, transdifferentiating into myofibroblasts and acquiring contractility and proinflammatory and fibrogenic properties [106]. Activated HSCs possess the ability to proliferate and migrate, thus accumulating in sites of tissue repair. These cells secrete large amounts of ECM and regulate its degradation, thus being able to degrade normal ECM and replace it with fibrous tissue. PDGF, mainly produced by Kupffer cells, is the predominant mitogen that promotes HSC activation. Collagen synthesis in HSCs is regulated at both transcriptional and post-transcriptional levels [109]. Increased stability of collagen mRNA is involved in increased collagen synthesis in activated HSCs. At the post-transcriptional level, collagen expression is regulated by sequences located in the 3′ region of the untranslated RNA, through the RNA-binding protein αCP2. Quiescent HSCs express markers characteristic of adipocytes (peroxisome proliferator activator receptor gamma (PPARγ), sterol regulatory element-binding protein 1 and leptin), and when activated they express myogenic markers (alpha-smooth muscle actin, c-myb and myocyte-specific enhancer factor 2A) [105]. In addition to HSCs, other liver cell types may also have fibrogenic potential. Myofibroblasts derived from portal vessel mesenchymal cells proliferate around the bile ducts in cholestasis-induced liver fibrosis to initiate collagen deposition [110]. The relative importance of each cell type in hepatic fibrogenesis may depend on the origin of the liver injury. While HSCs are the main fibrogenic cell type in pericentral areas, portal myofibroblasts may predominate when liver injury occurs around portal tracts [105].

Hepatocytes are the main hepatic parenchymal cells, and they play complex roles in fibrosis and cirrhosis. These cells are the target of most hepatotoxic agents, and chronic damage promotes their apoptosis and compensatory regeneration [111]. Damaged hepatocytes actively contribute to fibrogenesis, since they release ROS and fibrogenic mediators, thereby inducing the recruitment of inflammatory cells at the injured site and the activation of HSCs. In addition, there is evidence suggesting that hepatocytes can transform into fibroblasts through the so-called EMT. Zeisberg et al. demonstrated that by stimulating mouse hepatocytes with TGFβ1, the phenotype and functionality of the hepatocytes change, and there is also evidence that EMT occurs in vivo [112]. Finally, in late-stage fibrosis or cirrhosis, hypoxic hepatocytes become a predominant source of TGFβ1, further exacerbating liver fibrogenesis [113].

Progressive fibrosis with ECM deposition results from excessive synthesis mainly by HSCs and an imbalance in matrix degradation. The enzymes responsible for the correct homeostasis of the ECM are MMPs and their tissue inhibitors, TIMPs. Upon damage, upregulation of TIMPs occurs, resulting in inhibition of MMP activity and, consequently, matrix proteins accumulate in the extracellular space [114]. MMPs have complex regulation. In addition to being regulated by TIMPs, they are also regulated by different mechanisms at the transcription level (interleukin 1 α (IL1α), TGFβ, or tumor necrosis factor alpha (TNFα) cytokines) by the cleavage and activation of the pro-MMPs proenzyme, and by the action of inhibitors of endogenous proteinases, such as the α2-macroglobulins produced by hepatocytes [115,116,117]. In humans, 23 members of the MMP family have been described, although not all are expressed in liver tissue. Among the most studied in liver fibrosis are MMP1/MMP13, MMP2, MMP14, and MMP9. MMP13 is the rodent homolog to human MMP1. Macrophages are the main cellular source of MMP13/MMP1. On the other hand, four types of TIMPs have been described: TIMP1, TIMP2, TIMP3, and TIMP4. TIMPs bind to MMPs through their N-terminal domain. Each TIMP binds with higher binding strength to specific MMPs. TIMP1 binds to MMP9, while TIMP2 mainly inhibits MMP2. TIMP1 and TIMP2 play an important role in hepatic fibrosis. TIMP1 is mainly produced by HSCs, which constitute the major source of TIMPs in the fibrous liver. Early in the fibrosis phase, Kupffer cells and hepatocytes also contribute to TIMP1 production. Its transcription is mainly regulated by TGFβ, which in turn inhibits MMP1/MMP13, promoting the net accumulation of collagen. In addition to inhibiting the degradation of newly formed collagen fibrils by MMPs, TIMP1 can also prevent the activation of pro-MMPs [118]. TIMP2 is expressed in activated HSCs and Kupffer cells, but not in hepatocytes, and its expression increases with liver damage.

The first data suggesting that DLK1 could be involved in the pathogenesis of liver fibrosis were obtained by Huong et al. in 2004 [118]. These authors used a cytokine expression array and quantitative real-time PCR to study cytokine progression during liver fibrosis in biliary atresia. They found that DLK1 was upregulated during the early phase of biliary atresia and that the DLK1 protein was mainly, but not solely, present in HSCs. Moreover, DLK1 mRNA was present in hepatocytes. The authors concluded that DLK1 could be implicated in the activation of HSCs and in the fibrogenesis associated with biliary atresia. This hypothesis was later confirmed by Pan et al. [119] showed that hepatocytes may express DLK1 in response to injury signals and generate two soluble forms of DLK1. These authors proposed that the larger soluble form of 50 kDa is paracrined from the hepatocytes into HSCs, facilitating activation of HSCs and subsequent fibrogenesis. This was the first publication suggesting correlation between DLK1 expression and soluble DLK1 concentration in liver fibrosis. Later, investigations performed in partially hepatectomized mice also agreed with the concept that DLK1 was a major player in HSCs activation in liver regeneration [58], with this effect being exerted via epigenetic regression of PPARγ. These results were subsequently further supported by Pan et al. [120] on showing that low molecular weight FGF isoform treatment in mice with carbon tetrachloride (CCl_4_)-induced fibrosis resulted in attenuation of HSCs activation and fibrosis. This occurred by epigenetic down regulation of DLK1 expression through the p38 MAPK pathway. During liver injury, the loss of the histone mark H3K4me3 in the IG-DMR region of the *Dlk1-Dio3* locus causes the recruitment of the DNA methyltransferase 3a, which hypermethylates *Dlk1* and induces its overexpression. FGF2 recovers H3K4me3 levels, thereby preventing *Dlk1* upregulation and hepatic fibrogenesis.

The concept that PTTG1 could also be a regulator of liver fibrosis was almost simultaneously introduced by Buko W et al. [121]. These authors studied the development of liver fibrosis induced by thioacetamide in knockout and wild type *Pttg1* mice. The most important finding of this investigation was that thioacetamide-induced fibrosis development was significantly ameliorated in PTTG1 knockout mice. This was associated with diminished HSCs activation and suppressed circulating serum markers of inflammation, fibrosis, and angiogenesis. These results pointed to PTTG1 as a functionally required factor in the progression of liver fibrosis. Based on the concept that DLK1 is one of the most abundantly expressed PTTG1 targets [14], we recently documented the close link between DLK1 and PTTG1 in the progression of liver fibrosis [122]. In this investigation, *Pttg1* and *Dlk1* mRNA selectively increased in the liver of CCl_4_-treated rats and grew in parallel to fibrosis progression. Serum DLK1 concentrations correlated with hepatic collagen content and systemic and portal hemodynamics. In addition, human cirrhotic livers showed greater PTTG1 and DLK1 transcript abundance than non-cirrhotic human livers, and reduced collagen was observed in *Pttg1* knockout mice. The liver fibrotic molecular signature of these animals revealed lower expression of genes related to ECM remodeling. Furthermore, *Pttg1* silencing decreased the transcription of *Dlk1*, *collagens I* and *III*, and other transcripts involved in ECM remodeling. In summary, these findings further support the hypothesis that PTTG1/DLK1 signaling could be a novel pathway for targeting the progression of liver fibrosis.

### 2.3. Liver Cancer

HCC accounts for the second most common cause of cancer-related deaths among men worldwide [123]. HCC is an inflammation-related cancer which develops as a consequence of a long-standing multistep process in which a premalignant environment promotes the neoplastic transformation of hepatocytes [124]. Liver cirrhosis is the major risk factor for HCC, but up to 20% of HCC develops in non-cirrhotic patients. In most cases, liver transplantation and surgical resection are the only potential curative options, but HCC is often diagnosed in advanced stages of the disease when few therapeutic options are available [125].

Liver cirrhosis is the major risk factor for HCC. In short, sustained oxidative stress and non-resolved wound healing response induce lipid and protein oxidation, as well as genomic instability, which facilitate chromosomal aberrations and somatic gene mutations [126]. A broad spectrum of alterations has been found in HCC, but the most frequent driver gene mutation (around 60%) includes the gene encoding for telomerase reverse transcriptase, the enzyme that prevents telomere shortening during cell division. In the cirrhotic liver, telomerase activity is decreased, leading to hepatocyte senescence. However, during hepatocarcinogenesis, it is reactivated to enable uncontrolled proliferation of cancer cells [127]. High proliferation rates require elevated levels of energy expenditure and lead to the generation of a hypoxic environment which favors the transition from aerobic to anaerobic metabolism [128].

Hypoxia has also been related to the evasion of immune response by tumor cells. Macrophages and other immune cell are reeducated towards immune tolerance and further sustain tumor cell proliferation with, for instance, the production of mitogens [129]. Tumor-associated macrophages have a very important role in the modulation of the tumor microenvironment by promoting growth, invasion, and metastasis of tumor cells by secreting cytokines and mediators such as IL-10, IL-6, IL-8, TGFβ, VEGF, and MMPs, and activating intracellular signaling pathways including ERK, MAPK, and p38 [130,131].

Tumor cell cross talk with the microenvironment also induces EMT. It is a process in which epithelial cells lose their adhesive properties and polarity, acquire a mesenchymal migratory nature, augment the production of lysozymes involved in ECM degradation, and gain resistance to apoptosis [132]. Additionally, EMT can enhance the differentiation of transformed hepatocytes into cancer stem cells (CSCs), which can also be derived from bone marrow progenitors, thereby explaining the heterogeneity of tumor cells. These CSCs are believed to be cancer initiators, facilitate tumor development and growth, and thus invasion and metastasis of tumor cells [133,134].

The pathophysiology of HCC involves a complex multistep process. Therefore, there is an urgent need for better understanding of the underlying mechanisms involved in HCC initiation and progression in order to develop new biomarkers and treatment strategies to stop its progression.

PTTG1 is a protein involved in a myriad of functions, including cell cycle progression, DNA damage/repair, and apoptosis. It is overexpressed in numerous cancers but not in the adjacent non-cancerous tissue, including HCC. Besides, PTTG1 has been implicated in processes such as tumor induction, progression, and metastasis. Cho-Rok et al. [135] reported PTTG1 overexpression in several HCC cell lines as well as in human HCC cancer biopsies. Moreover, adenovirus-mediated silencing of PTTG1 resulted in the activation of p53 and induction of apoptosis in vitro and attenuated tumor cell growth in vivo when implanted in nude mice. Similar results were found by Liang et al. [136] in a study in which PTTG1 inhibition in human cancer-derived cell lines using a small interfering RNA (siRNA) also resulted in decreased proliferation and augmented apoptosis. Additionally, these authors reported a positive correlation between PTTG1 protein concentration and serum alpha-fetoprotein (AFP), portal vein tumor thrombosis, tumor stage, and FGF levels.

Given that T3 is implicated in growth regulation and that the liver is a well-known target for this hormone, Chen et al. [33] investigated its relationship with PTTG1 and the development of HCC. These authors found that in the liver, T3 indirectly reduces PTTG1 expression via suppression of Sp1-mediated PTTG1 promoter activity. In contrast, during HCC thyroid receptor expression is decreased and Sp1 and PTTG1 are augmented, promoting cell growth. Some years later, Lee et al. [8] suggested that the loss of the tumor suppressor KLF6 in HCC and the consequent activation of PTTG1 promoter activity, due to the loss of direct repression, could be another mechanism contributing to the increased proliferation, angiogenesis, and possibly chromosomal instability in HCC.

It has been estimated that hepatitis B virus (HBV) infection is responsible for at least 50% of HCC cases worldwide [137]. For this reason, several investigators have studied the potential role of PTTG1 in HBV-induced HCC. Molina-Jiménez et al. [19] demonstrated that PTTG1 expression is augmented in liver biopsies from patients with cirrhosis or HCC due to chronic HBV infection. This was due to prevention of PTTG1 ubiquitination and subsequent proteasomal degradation mediated by the HBV X protein. Li et al. [138] further showed that chronic infection with HBV induced the loss of miR-122, allowing viral replication by modulating p53 activity. As a result, miR-122 inhibition triggered the expression of its target gene PBF, the protein responsible for PTTG1 nuclear translocation and promotion of its transcriptional activity. PTTG1 translocation led to liver cancer cell proliferation, invasion, and tumor growth. More recently, Lin X et al. [139] performed a gene expression array comparing normal human liver, HCC-HBV infected and matched paracancer tissue specimens. They found that PTTG1 was the most upregulated gene in HCC and its expression positively correlated with that of TNFα. TNFα exposure to human-derived cancer cell lines caused PTTG1 induction. Moreover, siRNA-mediated *PTTG1* silencing resulted in downregulated c-myc expression and reduced proliferation. Interestingly, they found that PTTG1 correlated with more advanced tumor size and grade and was an independent predictor of poorer patient survival.

Other authors have specifically studied the relationship of PTTG1 upregulation with aggressive tumor behavior, poor prognosis, angiogenesis, and the development of metastasis. In 2006, Fujii et al. [140] suggested that *PTTG1* mRNA levels were upregulated in human HCC and correlated with serum AFP levels and intratumoral microvessel density. More importantly, *PTTG1* mRNA levels were a prognostic marker for disease-free and overall survival. Furthermore, El-Masry et al. [141] reported higher PTTG1 transcript levels in HCC metastatic cases compared to non-metastatic patients. More recently, using a bioinformatics analysis, Wang et al. [142] confirmed that the PTTG1 was one of the most upregulated genes in HCC and that its levels strongly correlated with patient prognosis and survival.

Zheng et al. [143] also showed that forkhead Box M1 (FoxM1) binds to the PTTG1 promoter at the −391 to −385 bp region and enhances its activity in colorectal cancer cells, promoting invasion and migration. Moreover, in vivo tumor xenograft studies, in which FoxM1 expressing wild type or knockout cells for PTTG1 were implanted in the spleen of nude mice, revealed decreased liver metastasis in the absence of PTTG1. Finally, the authors suggested that upregulated FoxM1-PTTG1 could modulate the TP53 pathway, leading to the suppression of the Wnt antagonist dickkopf homolog 1 and the enhancement of the Wnt pathway, which ultimately further induces PTTG1 transcription. Other mechanisms described for PTTG1 upregulation include the dysregulation of the long non-coding RNA PTTG 3 pseudogene. Huang et al. [144] suggested that PTTG3 upregulated PTTG1 and activated the PI3K/AKT signaling pathway, regulating genes related to cell cycle progression, apoptosis, and EMT. Finally, Yang et al. [134] reported that miR-374c-5p is downregulated in human HCC and is associated with poor prognosis and short survival. In the healthy liver, miR-374c-5p targets the 3′-untranslated region of PTTG1 to inhibit its expression. Therefore, its absence enhances PTTG1 expression and the consequent proliferation, EMT, migration, and invasion of HCC cancer cells.

During liver development, both membrane-bound and the large soluble form of DLK1 are upregulated in the precursors of hepatocytes and cholangiocytes, namely hepatoblasts [145]. According to the antagonist role of DLK1 and Notch, just prior birth DLK1 levels are downregulated and Notch1/Notch2 signaling is augmented enhancing hepatoblast maturation [37]. Huang et al. hypothesized that DLK1-drived hepatic stem cell differentiation is regulated by the AKT and MAPK pathways and FGF is an important mediator involved in this process [146]. While hepatic DLK1 levels dramatically drop to assure full liver maturation and correct bile duct development before birth, DLK1 has been reported to be expressed in a subpopulation of oval stem cells present in the rat adult liver [68]. Additionally, as previously described, during tumorigenesis some liver cells acquire stem like properties. For all these reasons, DLK1 has been suggested to be a potential biomarker in hepatoblastoma (HB), the primary malignant hepatic tumor in children.

After analyzing the gene expression patterns and global genomic alterations in human HCC and HB, Luo et al. [147] found that DLK1, insulin like growth factor 2, and paternally expressed 10 were the most upregulated genes in a subset of human HB compared to HCC samples. Accordingly, Dezso et al. [148] evaluated DLK1 protein expression in human HB, HCC, and non-cancerous surrounding tissue. In this group of patients, all HB expressed DLK1 in the epithelial components of the tissue. At the same time, Cairo et al. [149] performed a gene expression screening analysis to better understand the pathogenesis involved in HB development. The analysis of patient tumor samples revealed evidence of Wnt and β-catenin activation, as well as upregulation of several imprinted genes such as *DLK1* and *MEG3*. These authors also reported that tumor aggressiveness was associated with an interplay between Wnt and Myc. López-Terrada et al. [150] found that DLK1 was also strongly overexpressed in a mouse model of HB like c-myc induced tumors. No differences in the elevated DLK1 expression were found in the different HB histological subtypes. They speculated that HB could develop from a bipotential precursor, characterized by the expression of DLK1, and that the degree of Wnt or Notch pathway activation could depend on the stage of developmental arrest and therefore, be useful for tumor subtypes classification. Furthermore, Notch2 has been reported to be frequently overexpressed in HB, regulating cell growth and maintaining a population of undifferentiated hepatoblasts [151]. Therefore, a balance between DLK1 and Notch activation seems to be key in the development of HB [37].

More recently, Carrillo-Reixach et al. [152] performed a comprehensive genomic, transcriptomic, and epigenomic characterization to identify new biomarkers and therapeutic targets for HB. These authors found that Wnt/β-catenin pathway and imprinted and stem cell-related genes were upregulated in HB. They described an upregulation of the DLK1-DIO3 locus, and in particular, *DLK1*, *MEG3*, *SNORD113-3*, and *SNORD114-22* expression. This increase was related to activation and mutations in the Wnt/β-catenin pathway. HB was also associated with a general DNA hypomethylation and the degree of 14q32 locus expression correlated with the hypomethylation status of the region. Hovewer, the lowest levels of methylation were found in fetal liver samples. All these findings together suggest that overexpression of *DLK1-DIO3* locus genes is a hallmark of HB and strengthens the concept that HB recapitulates the pathological and molecular features of developing livers. In keeping with this, in the same year Honda S et al. [153] reported a dysregulation of microRNAs and small nucleolar RNAs (snoRNAs) located within the 14q32.2 *DLK1-DIO3* imprinted region due to an overall loss of DNA methylation in metastatic HB.

Additionally, variable DLK1 protein levels of between 4 and 57% have been reported in patient HCC liver biopsies [50]. Huang et al. [154] found elevated expression of the DLK1 gene and protein, measured by real-time PCR and immunohistochemistry, respectively, in human samples of HCC. In some of these specimens, DLK1 upregulation was associated with demethylation of DLK1 promoter and hypermethylation of the IG-DMR. Suppression of endogenous DLK1 expression resulted in decreased cell growth, colony formation, and tumorigenicity in human-derived liver cancer cell lines. Given the fact that not all HCC strongly express DLK1, Jin et al. [155] evaluated the prognostic value of DLK1. These authors found that DLK1 positive tumors were associated with a shorter survival time compared to negative tumors. DLK1 presence did not correlate with other classic prognostic factors such as AFP, tumor node metastasis, or vascular invasion. Some years later, Yanai et al. [156] also examined DLK1 protein expression in normal adult liver, in non-neoplastic liver diseases (viral hepatitis and nodular cirrhosis), as well as in neoplastic lesions (intrahepatic cholangioma and HCC). Positive staining was found only in some HCC specimens, and it was more frequent in patients under 50 years old and in AFP positive carcinomas. Despite the exact mechanism being unknown, disparity in DLK1 expression between different HCC specimens could partially explained by Notch pathway modulation. Some authors suggested that Notch1 could act as a tumor suppressor, whereas others reported Notch as a driver for hepatocyte dedifferentiation and tumor progression [157]. Luk et al. [158] found out that in human and mouse HCC, the *DLK1-DIO3* locus displayed a change in the imprinting status. This led to upregulation of the miRNAs cluster as well as the expression of imprinted gene transcripts including *DLK1*, especially in a subset of patients with poor clinical outcomes. Finally, the dysregulation correlated with Wnt/β-catenin signaling downstream targets, which are implicated in CSC activation. Almost in parallel, Lempiäinen et al. [159] confirmed the involvement of β-catenin signaling in the regulation of *DLK1-DIO3* cluster expression in HCC.

Further insight on potential of DLK1 as a biomarker for HCC CSC was described by Xu et al. [160]. These authors assessed the expression of DLK1 positive cells in some of the human cancer-derived cell lines most commonly used in laboratories. They showed that only a small subpopulation of these cell lines was positive for DLK1 as detected by flow cytometry. In addition, this subpopulation was enriched in spheroid colonies, co-expressed markers of progenitor cells such *Nanog* or *Sox2*, and displayed potential of self-renewal and possessed higher chemoresistance. Li et al. [161] also evaluated the predictive value of serum DLK1 levels as a prognostic biomarker in HCC. Patients with HCC presented increased DLK1 levels compared to healthy subjects of patients with chronic HBV infection. No differences were found between patients with HCC or HBV-derived cirrhosis. Furthermore, serum DLK1 correlated with AFP and tumor size in HCC patients. Finally, shorter survival was found in patients expressing elevated DLK1 levels.

Several years later, Cai et al. [162] suggested that targeting DLK1 expression could promote CSC differentiation and produce an anti-tumoral effect. They showed that DLK1 silencing reduced tumor growth in an orthotropic xenograft as well as in a chemical DEN-HCC induced model due to a G1 phase arrest. Although the underlying molecular mechanism was not deciphered, they proposed that DLK1 inhibition in HCC might induce CSC differentiation. Liem et al. [163] further supported the idea that DLK1 could promote cancer cell stemness and tumorigenicity via the modulation of Wnt and PPARγ pathways. They suggested that the hypoxic microenvironment of the tumor could induce DLK1 expression, a positive regulator of Wnt signaling. Wnt/β-catenin pathway is activated in up to 50% of HCC and regulates numerous processes related to tumor initiation, growth, survival, migration, differentiation, and apoptosis [164]. Additionally, DLK1 has been shown to suppress PPARγ, a widely known tumor growth, invasion, and EMT inhibitor. Finally, Seino et al. [165] evaluated the expression of several liver progenitor cell markers in HCC and their prognostic potential. DLK1 was the most upregulated marker evaluated and correlated with worse free survival. In agreement with previous studies, DLK1 correlated with AFP levels.

In conclusion, there are substantial data strongly suggesting that the activation of the PTTG1/DLK1 axis is a key factor involved in the pathophysiology of NAFLD, liver fibrosis, and liver cancer, which, in turn, raises the hypothesis that disruption of this signaling pathway could be of therapeutic utility in these pathological conditions.

## Figures and Tables

**Figure 1 ijms-23-06897-f001:**
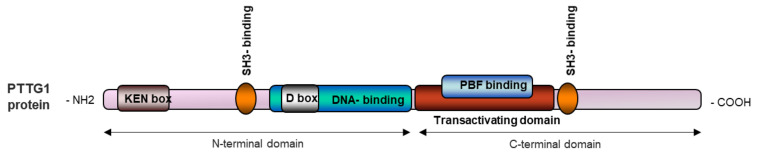
Schematic representation of the PTTG1 protein.

**Figure 2 ijms-23-06897-f002:**
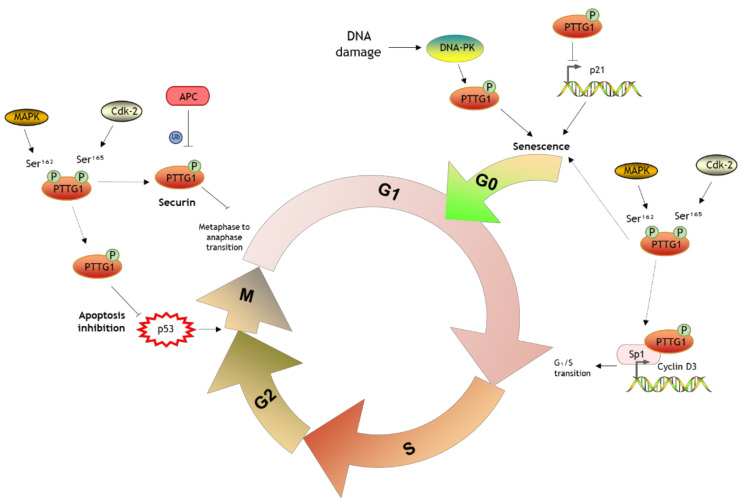
Schematic representation of the role of PTTG1 in cell cycle regulation.

**Figure 3 ijms-23-06897-f003:**
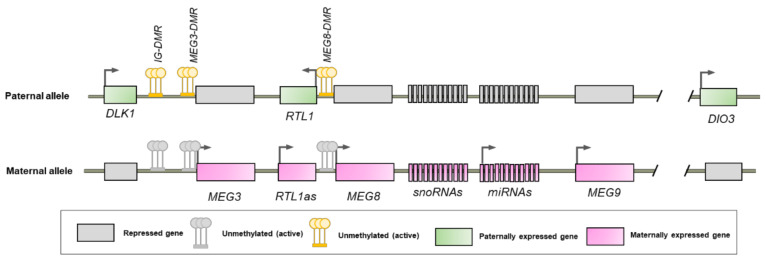
Schematic representation of the maternal and paternal alleles of the DLK1-DIO3 cluster.

**Figure 4 ijms-23-06897-f004:**
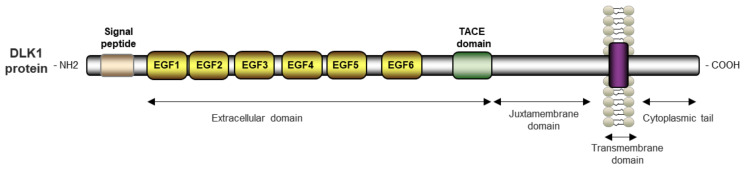
Schematic representation of human DLK1 protein.

**Figure 5 ijms-23-06897-f005:**
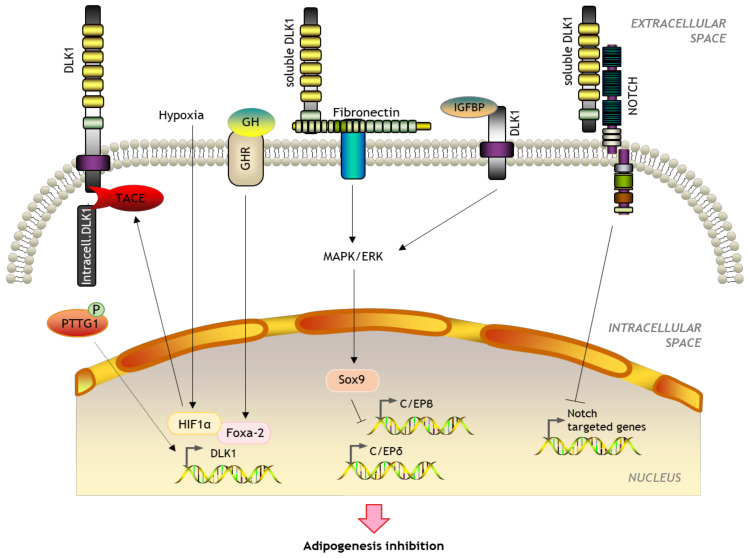
Schematic representation of the role of DLK1 in adipogenesis.

## Data Availability

Not applicable.

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
