# Peer review of "Pituitary Tumor-Transforming Gene 1/Delta like Non-Canonical Notch Ligand 1 Signaling in Chronic Liver Diseases"

_ijms, 2022, doi:10.3390/ijms23136897_

Round 1
Reviewer 1 Report
Reviewer’s comments
This review article primarily focuses on the roles of PTTG1/DLK signaling in the pathogenesis of chronic liver disease. It seems to be very interesting. I deeply appreciate your efforts. However, several revisions are required to improve the quality of the article. Please refer to the comments shown below.
Major
#1. The authors should select appropriate key words to reflect the content of the review article. Hepatic steatosis, hepatic fibrosis and hepatocarcinogenesis should be added.
#2. This article primarily focuses on the roles of PTTG1/DLK signaling in the following three steps: hepatic steatosis, hepatic fibrosis and hepatocarcinogenesis. The authors should illustrate what PTTG1/DLK may affect in each step more clearly. The statement in the graphic abstract is insufficient.
#3. The authors should describe whether or not hepatic expressions of PTTG1 and/or DLK1 are upregulated in the experimental models of NASH. If they are upregulated, where are they predominantly upregulated in the liver? (hepatoctytes? Kupffer cells? HSCs?). The localization of PTTG1 and/or DLK1 is extremely important in the pathogenesis of chronic liver disease.
#4. The efficacy of soluble DLK1 has been shown to induce NOTCH-dependent and –independent signaling pathway (lines275-276). However, its efficacy in the role of hepatic steatosis, hepatic fibrosis and hepatocarcinogenesis has not. The authors should appropriately mention the efficacy of soluble DLK1 levels in these phenomena.
#5. The explanation for Figure 6 seems to be insufficient. The authors should explain the figure more clearly.
#6. The authors should introduce the candidates for molecular targets to inhibit the binding of PTTG1 to DLK1 if the molecular targets have been identified.
Minor
#1. HIF-1 should be corrected to HIF-1a (line 54).
#2. Cyclin-dependent kinase-2 should be abbreviated as Cdk-2 (line 125).
#3. snRNA should be spelled out as small nucleolar RNA (line 655).
#4. The numbering of the references is wrong from the reference #91. (see line 905)
Author Response
Major
#1. The authors should select appropriate key words to reflect the content of the review article. Hepatic steatosis, hepatic fibrosis and hepatocarcinogenesis should be added.
- Following the reviewer’s suggestion, the keywords ¨hepatic steatosis, hepatic fibrosis and hepatocarcinogenesis¨ have been included in the new version of the review.
#2. This article primarily focuses on the roles of PTTG1/DLK signaling in the following three steps: hepatic steatosis, hepatic fibrosis and hepatocarcinogenesis. The authors should illustrate what PTTG1/DLK may affect in each step more clearly. The statement in the graphic abstract is insufficient.
- In the current review, we have performed an extensive bibliographic search addressing the topic. Unfortunately, the exact role of PTTG1 and DLK1 signaling in these different steps is not yet completely defined. However, in an attempt to clarify this issue, we have enlarged the graphical abstract in agreement with the suggestion of the reviewer.
#3. The authors should describe whether or not hepatic expressions of PTTG1 and/or DLK1 are upregulated in the experimental models of NASH. If they are upregulated, where are they predominantly upregulated in the liver? (hepatoctytes? Kupffer cells? HSCs?). The localization of PTTG1 and/or DLK1 is extremely important in the pathogenesis of chronic liver disease.
- As far as we know there are no studies directly assessing DLK1 or PTTG1 hepatic levels in experimental models of NASH. Indirect evidence of decreased DLK1 expression in metabolic diseases has been given by several authors via overexpression of DLK1. For instance, Villena et al. (98) reported that transgenic mice overexpressing DLK1 regulated by an adipose specific promoter were resistant to diet-induced obesity. More recently, Charalambous et al. (99) showed that DLK1 overexpression decreased hepatic lipogenesis and protected against diet-induced steatosis in mice fed a high-fat diet. Evaluation of PTTG1 and DLK1 expression in the liver and different cell types has been assessed in experimental models of chronic liver diseases other than NASH, including fibrosis and hepatic regeneration (118, 119, 122).
#4. The efficacy of soluble DLK1 has been shown to induce NOTCH-dependent and –independent signaling pathway (lines275-276). However, its efficacy in the role of hepatic steatosis, hepatic fibrosis and hepatocarcinogenesis has not. The authors should appropriately mention the efficacy of soluble DLK1 levels in these phenomena.
- Given the importance of NOTCH signaling in CLD development and the antagonism of DLK1 to NOTCH signaling, we have extended and clarified its role in NASH (page 10, lines 395-402 and 404-406) and liver cancer (page 15 lines 640-643 and 668-671 and page 16 lines 702-706)
#5. The explanation for Figure 6 seems to be insufficient. The authors should explain the figure more clearly.
- According to the comments of both reviewers, figure 6 has been removed from the new version of the manuscript.
#6. The authors should introduce the candidates for molecular targets to inhibit the binding of PTTG1 to DLK1 if the molecular targets have been identified.
- As far as we know, direct binding of PTTG1 to DLK1 has not been shown. Espina et al. (14) attempted to describe the mechanism underlying PTTG1-mediated DLK1 upregulation. However, they could not demonstrate that PTTG1 was a transcriptional regulator of DLK1 since they did not find PTTG1 response elements in the Dlk1 promoter regions assessed (191, 1400, and 4062 base pairs upstream of the transcriptional start site). Nonetheless, they could not exclude the existence of PTTG1 response elements in other DLK1 gene regions. Additionally, these authors also showed that PTTG1 does not interact with DLK1 mRNA to increase their stability. Therefore, they suggested that one or more unknown proteins could be involved in PTTG1-mediated DLK1 upregulation.
Minor
#1. HIF-1 should be corrected to HIF-1a (line 54).
#2. Cyclin-dependent kinase-2 should be abbreviated as Cdk-2 (line 125).
#3. snRNA should be spelled out as small nucleolar RNA (line 655).
- All these changes and corrections have been made in the revised article.
#4. The numbering of the references is wrong from the reference #91. (see line 905)
- Reference #92 was mixed up with reference #91 and #94 with #93. We have corrected these editing mistakes in the new version of the manuscript.
Reviewer 2 Report
The authors present an extensive and comprehensible review on the PTTG1/DLK1 axis. All major scientific progress concerning the understanding of its involvement in the progress of liver disease progression up to HCC development is included. The citations are up to date. The manuscript is well organized with a coherent structure.
- lines 12-13: Liver steatosis, liver fibrosis and liver cancer are different and are sometimes concomitant manifestations of chronic liver diseases (CLDs). - this needs to be rephrased. Steatosis, fibrosis and HCC are not 'sometimes' concomitant but represent different steps in CLD progression and have overlapping hallmarks. This first sentence appears unworthy regarding the good quality of the rest of the review manuscript
lines 13-14: The management of CLDs remains a challenge and identifying effective treatment is a major unmet medical need. - This way to unspecific. CLD is to diverse to identify one effective treatment. Please rephrase.
-Figure 1: lettering is to small
- lines 335-337: The abundance of underlying disease of CLD differs immensely between different regions. Please clarify.
- Figure 6: I do not see the point of showing that picture since no immediate connection to the main subject of this review is drawn. Erase or replace by a figure including a immediate connection to PTTG1/DLK1
- lines 516 - 521: please rewrite. Up to 20% of HCCs develop in non-cirrhotic patients, thus not ALL HCCs necessarily develop in a premalignant environment. Furthermore, surgical resection is a curative treatment option in these non-cirrhotic patients and liver transplantation is not always "the only" curative option, eventhough, of course, in the majority of cases this is true
I suggest acceptance of this good review article after adressing minor revisions and moderate language and style improvements
Author Response
1# lines 12-13: Liver steatosis, liver fibrosis and liver cancer are different and are sometimes concomitant manifestations of chronic liver diseases (CLDs). - this needs to be rephrased. Steatosis, fibrosis and HCC are not 'sometimes' concomitant but represent different steps in CLD progression and have overlapping hallmarks. This first sentence appears unworthy regarding the good quality of the rest of the review manuscript
- Following the indications of the reviewer, we have rephrased the sentence.
2# lines 13-14: The management of CLDs remains a challenge and identifying effective treatment is a major unmet medical need. - This way to unspecific. CLD is too diverse to identify one effective treatment. Please rephrase.
- The sentence has been modified in line with the reviewer’s
3# Figure 1: lettering is too small
- The size of lettering in figure 1 has been increased.
4# lines 335-337: The abundance of underlying disease of CLD differs immensely between different regions. Please clarify.
- We agree with the reviewer’s observation. To clarify this, the following sentence has been added to the new version of the review: “The global burden of CDLs is rising every year. However, its prevalence varies among regions largely due to differences in the incidence of risk factors including obesity, viral hepatitis and alcohol consumption. Therefore, demographic and historical factors need to be taken into account when trying to understand the epidemiology of CDLs.” (page 9).
5# Figure 6: I do not see the point of showing that picture since no immediate connection to the main subject of this review is drawn. Erase or replace by a figure including an immediate connection to PTTG1/DLK1
- As suggested by both reviewers, figure 6 has been omitted from the new version of the article.
6# lines 516 - 521: please rewrite. Up to 20% of HCCs develop in non-cirrhotic patients, thus not ALL HCCs necessarily develop in a premalignant environment. Furthermore, surgical resection is a curative treatment option in these non-cirrhotic patients and liver transplantation is not always "the only" curative option, eventhough, of course, in the majority of cases this is true
- The sentence has been rewritten, as suggested by the reviewer.
I suggest acceptance of this good review article after addressing minor revisions and moderate language and style improvements
Round 2
Reviewer 1 Report
Reviewer’s comments (2nd round)
The authors responded very well to the reviewer’s comment except for #2 and #4. Please refer to the comments shown below.
#2. The activation of DLk1 signaling is responsible for the progression to NASH from NAFL (simple steatosis) as shown in Graphical abstract. However, a previous study revealed that an increase in DLK1 gene expression resulted in a reduction of hepatic steatosis by inhibiting Notch signaling. This seems to be paradoxical. Moreover, what commands to upregulate or downregulate DLK1 expression? Please describe authors’ opinion.
#4. I would like to know whether or not soluble DLK1 levels were significantly associated with DLK1 expression in hepatic steatosis, hepatic fibrosis and hepatocarcinogenesis. Please mention the correlation and soluble DLK1 levels and DLK1 expression in each process.
Minor
I found several parts which require minor revisions in the text.
#1. NOTCH and Notch were mixed in the text. They should be united as NOTCH or Notch.
#2. Insulin growth factor (IGF-1) should be corrected to “insulin-like growth factor (IGF-1) (page 5, line 166).
#3. IGF-1 is synthesized in the hepatocytes. “Pituitary insulin-like growth factor-1” is wrong (page 10, line 392).
Author Response
Reviewer’s comments (2nd round)
Major
#2. The activation of DLk1 signaling is responsible for the progression to NASH from NAFL (simple steatosis) as shown in Graphical abstract. However, a previous study revealed that an increase in DLK1 gene expression resulted in a reduction of hepatic steatosis by inhibiting Notch signaling. This seems to be paradoxical. Moreover, what commands to upregulate or downregulate DLK1 expression? Please describe authors’ opinion.
- DLK1 has classically been described as an adipogenesis inhibitor but as the reviewer has suggested, contradictory roles of DLK1 in NAFLD progression have been found by several authors. Some investigators associate DLK1 upregulation to increased body fat and insulin resistance (101), while others indicate that DLK1 overexpression protects from obesity (98, 100). Moreover, DLK1 has a complex regulation and it interferes with a myriad of different signaling pathways including Notch, growth hormone and insulin-like growth factor. Therefore, the exact role of DLK1 remains elusive and controversial. As far as we know, more investigations addressing this issue need to be performed given that:
- Most studies addressing DLK1 role in adipogenesis use in vitro cell lines.
- There is need of studies directly evaluating serum soluble DLK1 and DLK1 mRNA expression in hepatic tissue of NAFLD patients or experimental animal models.
- Probably the effect of soluble DLK1 in obesity/fat accumulation is not the same in the different tissues including white adipose tissue, liver, skeletal muscle and pancreas.
- Not all studies use similar approaches to overexpress DLK1 or to measure its effects in fat deposition.
#4. I would like to know whether or not soluble DLK1 levels were significantly associated with DLK1 expression in hepatic steatosis, hepatic fibrosis and hepatocarcinogenesis. Please mention the correlation and soluble DLK1 levels and DLK1 expression in each process.
- To our knowledge there are no data regarding DLK1 serum levels and DLK1 mRNA expression in NAFLD. Indirect evidence of DLK1 regulation in NAFLD has been obtained by several experimental models but its exact role remains elusive and controversial. Villena JA et al. (98) demonstrated that transgenic mice overexpressing DLK1 protein were protected from fat-diet induced-obesity but showed aggravated insulin resistance and increased circulating lipid levels. In line, Kavalkova P et al. (99) reported that normal-weight or obese female human subjects did not show differences in serum DLK1 concentrations, whereas females with obesity and type 2 diabetes mellitus displayed significantly increased levels of circulating DLK1 compared to normal-weight subjects. Moreover, Lee YH et al. (103) showed that the administration of soluble DLK1 improved insulin tolerance and decreased hepatic triglycerides and lipid droplets in the liver of db/db mice. In contrast, Jensen CH et al. (101) found that elevated DLK1 circulating levels were associated to increased body fat and the development of metabolic syndrome in humans. They also showed that DLK1 deficiency protected against obesity and insulin resistance by negatively regulating GLUT4-mediated skeletal muscle glucose uptake in a high-fat diet experimental mice model. In order to make clearer the involvement of DLK1 in obesity/NAFLD we have included more studies regarding serum DLK1 levels in the new version of the review (page 10). Regarding hepatic fibrosis, Pan RL et al. (120) reported that DLK1 expression was induced after liver injury first in hepatocytes and then in a paracrine manner, in hepatic stellate cells. The larger 50 kDa soluble form being the responsible for hepatic stellate cells activation. We have also reported (124) that DLK1 hepatic mRNA expression selectively and gradually increases with progression of experimental liver fibrosis and reaches the highest levels in cirrhosis. Our study also showed that this phenomenon is paralleled by increasing serum levels of the large soluble 50 kDa DLK1 fragment. Furthermore, we found significantly elevated hepatic DLK1 mRNA expression in cirrhotic human livers as compared to non-cirrhotic livers. Finally, DLK1 expression has also been associated to hepatocarcinogenesis. Li H et al. (163) reported augmented serum DLK1 levels in patients with HCC compared to healthy individuals. Besides, Huang J et al. (156) found upregulated hepatic DLK1 protein and mRNA in HCC specimens.
Minor
I found several parts which require minor revisions in the text.
#1. NOTCH and Notch were mixed in the text. They should be united as NOTCH or Notch.
#2. Insulin growth factor (IGF-1) should be corrected to “insulin-like growth factor (IGF-1) (page 5, line 166).
#3. IGF-1 is synthesized in the hepatocytes. “Pituitary insulin-like growth factor-1” is wrong (page 10, line 392).
- All these changes have been applied in the new version of the manuscript.

Round 3
Reviewer 1 Report
Reviewer’ comments (3rd round)
The authors appropriately responded to the reviewer’s comment #2. However, the reply to the comment #4 was not sufficient.
The statement on the correlation between DLK1 expression and soluble DLK1 concentration is .quite essential. According to the previous study by Pan et al (#120), “hepatocytes may express DLK1 in response to injury signals and generate two soluble forms of DLK1. The larger soluble form was paracrined from the hepatocytes into HSCs, facilitating activation of HSCs and subsequent fibrogenesis.” Of course, Pan et al. did not provide the direct evidence on the correlation. However, they speculated the correlation in detail. The authors should refer the statement.
Author Response
Reviewer’s comments (3rd round)
Major
The authors appropriately responded to the reviewer’s comment #2. However, the reply to the comment #4 was not sufficient.
The statement on the correlation between DLK1 expression and soluble DLK1 concentration is quite essential. According to the previous study by Pan et al (#120), “hepatocytes may express DLK1 in response to injury signals and generate two soluble forms of DLK1. The larger soluble form was paracrined from the hepatocytes into HSCs, facilitating activation of HSCs and subsequent fibrogenesis.” Of course, Pan et al. did not provide the direct evidence on the correlation. However, they speculated the correlation in detail. The authors should refer the statement.
- According to the reviewer comment, we have referred to this statement in the page 12 of the re-reviewed manuscript.
